# Spheroid Culture System, a Promising Method for Chondrogenic Differentiation of Dental Mesenchymal Stem Cells

**DOI:** 10.3390/biomedicines11051314

**Published:** 2023-04-28

**Authors:** Caroline Mélou, Pascal Pellen-Mussi, Solen Novello, Damien Brézulier, Agnès Novella, Sylvie Tricot, Pascale Bellaud, Dominique Chauvel-Lebret

**Affiliations:** 1CNRS, ISCR (Institut des Sciences Chimiques de Rennes), University of Rennes, UMR 6226, 35000 Rennes, France; 2Pôle d’Odontologie, Centre Hospitalier Universitaire de Rennes, 35033 Rennes, France; 3UFR Odontologie, University of Rennes, 35043 Rennes, France; 4CNRS, Inserm UMS Biosit, France BioImaging, Core Facility H2P2, University of Rennes, 35000 Rennes, France

**Keywords:** cellular spheroid, chondrocyte, tridimensional cell culture, chondrogenic differentiation, mesenchymal stem cells

## Abstract

The objective of the present work was to develop a three-dimensional culture model to evaluate, in a short period of time, cartilage tissue engineering protocols. The spheroids were compared with the gold standard pellet culture. The dental mesenchymal stem cell lines were from pulp and periodontal ligament. The evaluation used RT-qPCR and Alcian Blue staining of the cartilage matrix. This study showed that the spheroid model allowed for obtaining greater fluctuations of the chondrogenesis markers than for the pellet one. The two cell lines, although originating from the same organ, led to different biological responses. Finally, biological changes were detectable for short periods of time. In summary, this work demonstrated that the spheroid model is a valuable tool for studying chondrogenesis and the mechanisms of osteoarthritis, and evaluating cartilage tissue engineering protocols.

## 1. Introduction

Articular cartilage is an avascular tissue with very low self-repair potential after trauma or degenerative disease [1,2]. This limited regeneration capacity can lead to degenerative changes in the traumatized cartilage, which represents a risk factor for the early development of osteoarthritis (OA) [3].

Different strategies have been developed to treat cartilage injuries, such as autologous chondrocyte implantation, microfracture, and mosaicplasty, but they have disadvantages such as donor site morbidity (and consequent osteoarthritic changes in the donor joint) and/or inferior repair as fibrocartilage [1,3,4,5].

Tissue engineering, which aims to restore, maintain, and improve tissue performance by combining stem cells and biomaterial scaffolds, could promote cartilage repair [6,7].

Regarding stem cells, in recent years, Mesenchymal Stem Cells (MSCs) have emerged as a promising cell source for the treatment of various degenerative, inflammatory, and autoimmune diseases [5]. Nevertheless, the therapeutic effects of MSCs are increasingly being attributed to paracrine secretion, particularly to nanovesicles called exosomes [8]. Investigation regarding the use of MSC exosomes as cell-free alternatives to MSCs for tissue engineering, particularly bone and cartilage tissue engineering, is increasing [9,10]. Thus, a tissue engineering technique which combines a bioactive glass scaffold with MSC exosomes could promote cartilage repair [11,12].

MSCs are multipotent cells that can differentiate into multiple cell lineages including osteoblasts, chondrocytes, adipocytes, endothelial cells, muscle cells, and neurons [2,13,14,15,16].

In the mentioned context, MSCs have a twofold interest:First, due to their multilineage differentiation potential, they could be differentiated into chondrocytes [1,2].Secondly, their exosomes could be used for cartilage repair [5].

Further, MSCs are known to exert immunomodulatory and anti-inflammatory effects [17]. MSCs can be isolated from various tissues (bone marrow, adipose tissue, umbilical cord, skeletal muscle, cartilage, synovium, cardiac tissue, and dental tissue) [18,19,20,21,22] and MSCs differentiation can be influenced by the cell source [23].

Bone marrow MSCs are the most extensively investigated MSCs and are considered as the gold standard. Nevertheless, obtaining them is often painful and carries the risk of infection [13].

Dental MSCs could be the solution to overcome this disadvantage. They are have a very accessible source since they can be isolated during routine dental surgeries [13]. Besides, they have many benefits:They are expandable and have relative genomic stability for a long period of time [18].They can differentiate into multiple lineages including odontoblasts, osteoblasts, chondrocytes, myocytes, neurocytes, adipocytes, corneal epithelial cells, and melanocytes [19].The mean doubling time frequency of dental pulp MSCs is comparable to that of bone marrow MSCs [13].The frequency of colony-forming cells from dental pulp is high compared to those from bone marrow [13].They have demonstrated immunomodulatory properties due to secreting cytokines [18].They have been found in various dental tissues (dental pulp, apical papilla, periodontal ligament, gingiva, dental follicle, tooth germ, and alveolar bone) [18].

All of these features make dental MSCs distinct from the other human MSCs and an effective tool for stem cell therapy [18].

The aim of this study was to develop an effective strategy to differentiate dental MSCs into chondrocytes, using three-dimensional cell cultures as spheroids. Our laboratory studies silicate bioactive glasses in bone and cartilage tissue engineering. Several recent studies have demonstrated the role of bioactive glasses in chondrogenesis and cartilage repair [12,24]. These are, therefore, relevant candidates for osteoarthritis. The objective of this study was to develop a three-dimensional chondrocyte culture model for screening innovative bioactive glasses to evaluate our cartilage tissue engineering experiments and for a mechanobiology model. For this last point, we demonstrated that spheroids were valuable tools for assessing mechanical stress [25].

For this purpose, the chondrogenic differentiation potential of MSCs from two different sources (dental pulp and periodontal ligament) was assessed.

The chondrogenic potential of MSCs cultured in spheroids was compared with that of MSCs cultured in monolayer (negative control) and pellets, pellet cultures being another type of tridimensional cell culture, which is the gold standard for maintaining chondrocytes in a differentiated stage [26].

## 2. Materials and Methods

In this study, two dental cell types were compared: human Dental Pulp Stem Cells (hDPSC) and human Periodontal Ligament Stem Cells (hPDLSC).

### 2.1. Isolation and Culture

hDPSC were obtained by explant culture. Extracted third molars were collected from healthy donors in agreement with French legislation (informed patients and Institutional Review Board approval/Registration number: DC-2012-1573, 19 April 2012). The pulp tissue was extracted from the teeth in an aseptic way, rinsed with Minimum Essential Medium–Alpha Eagle (α-MEM, Lonza, Verviers, Belgium), and explanted in a 35 mm Petri dish at seven 1 mm^3^ pieces/well. Explants were cultured in a complete culture medium consisting of α-MEM supplemented with 10% fetal bovine serum (FBS, Gibco, Life Technologies, Paisley, UK), 2 mmol/L glutamine (Lonza, Verviers, Belgium), 20 mmol/L HEPES (Lonza, Verviers, Belgium), and antibiotics (100 U/mL penicillin, 100 μg/mL streptomycin, Lonza, Verviers, Belgium) until cells emerged from the explant. These cells were cultured at 37 °C in a humidified atmosphere of 5% CO_2_.

hPDLSC obtained in the same way in a previous study were used in the present work [27]. Cells were used from passages 3 to 7.

### 2.2. Characterization

hPDLSC were characterized in a previous study: their MSCs-like characteristics were confirmed (spindle appearance, expression of the markers CD73, CD90, and CD105, no expression of the markers CD34 and CD45, capacity of in vitro differentiation into adipocyte and osteoblast lineages) [27].

The characterization of hDPSC was performed in the same way: first, a phenotypic characterization was done by flow cytometry to investigate the expression of the surface adhesion markers.

Then, a functional characterization was performed:To confirm the stemness of the hDPSC, their ability to form colonies was assessed with the Colony-Forming Assay.Then, to ensure that the hDPSC were still functional, their capacity for in vitro differentiation into adipocyte and osteoblast lineages was evaluated.

#### 2.2.1. Phenotypic Analysis by Flow Cytometry

Immunophenotypic characterization was performed by flow cytometric analysis. hDPSC were harvested and washed with phosphate buffer saline (PBS) (Lonza, Verviers, Belgium). BV421 mouse anti-human CD34, FITC mouse anti-human CD45, PE mouse anti-human CD73, PE-Cy7 mouse anti-human CD90, and APC mouse anti-human CD105 were incubated with cells for 15 min in darkness at room temperature (RT). Then, cells were washed with PBS for further analysis. A BD LSRFortessa™ X-20 flow cytometer system (BD Biosciences, San Jose, CA, USA) was used for antigen expression. Antibodies and buffers were purchased from BD Biosciences (San Jose, CA, USA).

#### 2.2.2. Functional Characterization

##### Colony-Forming Assay

hDPSC were seeded in 6-well plates on the basis of 500 cells/well and cultured in complete medium. On day 10, they were fixed for 30 min at RT with formaldehyde (Sigma-Aldrich, Saint-Louis, MO, USA) 3.7% in PBS. Crystal violet (0.5% in methanol) was used for 5 min and then samples were washed with distilled water [28]. Colony formation was observed.

##### Osteogenic Differentiation

To study osteoblastic differentiation, cells were seeded in 6-well plates (5 × 10^4^ cells/mL). On day 1, the medium was replaced by an osteoinductive one supplemented with 50 μg/mL of ascorbic acid (Sigma-Aldrich A4403, Saint-Louis, MO, USA), 10 mM of β-glycerophosphate (Sigma-Aldrich G9422, Saint-Louis, MO, USA), and 100 nM of dexamethasone (Sigma-Aldrich D4902, Saint-Louis, MO, USA). The supernatant was changed every 2 or 3 days. On day 21, cells were fixed with formaldehyde 10% for 15 min at RT, stained with 40 mM Alizarin Red solution (Sigma-Aldrich A5533, Saint-Louis, MO, USA) for 20 min at RT, and washed with distilled water.

##### Adipogenic Differentiation

To assess adipocyte differentiation potential, cells were seeded in 6-well plates (5 × 10^4^ cells/mL). On day 1, the medium was replaced by an adipocyte differentiation medium supplemented with 100 μM indomethacin (Sigma-Aldrich I7378, Saint-Louis, MO, USA), 0.5 mM 3-isobutyl-1-methylxanthine (Sigma-Aldrich I5879, Saint-Louis, MO, USA), 1 μM dexamethasone (Sigma-Aldrich D4902, Saint-Louis, MO, USA), and 1 μM insulin (Sigma-Aldrich I1882, Saint-Louis, MO, USA). Medium was changed every 2 or 3 days. On day 21, cells were fixed with formaldehyde 10% for 30 min at RT, stained with Oil Red O solution diluted at 60% in distilled water (Sigma-Aldrich O1391, Saint-Louis, MO, USA) for 30 min at RT, and washed with distilled water.

### 2.3. Chondrogenic Differentiation

For chondrogenic differentiation assessment, hDPSC and hPDLSC were both cultured under 6 experimental conditions: monolayer, pellets, and spheroids, in complete medium and in chondrogenic medium (Figure 1).

Chondrogenic medium used was Dulbecco’s modified Eagle’s medium (DMEM, Lonza, Verviers, Belgium) supplemented with 10% fetal bovine serum, 2 mmol/L glutamine, 20 mmol/L HEPES, antibiotics (100 U/mL penicillin, 100 μg/mL streptomycin), 1 mM sodium pyruvate (Lonza, Verviers, Belgium), 50 µg/mL ascorbic acid, 0.1 µM dexamethasone, 1% insulin-transferrin-selenium (ITS, Sigma-Aldrich I3146, Saint-Louis, MO, USA), and 10 ng/mL TGFβ3 (Sigma-Aldrich SRP3171, Saint-Louis, MO, USA).

Cells were seeded at 2.5 × 10^4^ cells/mL for monolayer, 5 × 10^3^ cells per spheroid, and 1.25 × 10^5^ cells per pellet.

Spheroids were obtained using the liquid overlay technique, as previously described [29]. Briefly, 96-well plates were treated with 1% agarose prepared in PBS to form a non-adhesive surface. Cells were detached from culture flasks by trypsin/EDTA and a single cell suspension was prepared at 2.5 × 10^4^ cells/mL. To initiate spheroid formation, 200 μL were seeded into individual wells and incubated at 37 °C.

Pellets were obtained in a v-bottom 96-well plate, as previously described [30]; cells were seeded and centrifugated at 355× *g* for 5 min.

#### 2.3.1. RNA Extraction and Real-Time Quantitative Polymerase Chain Reaction (RT-qPCR)

On days 3 and 7, total RNA extraction from hDPSC and hPDLSC was performed using a Nucleospin RNA extraction kit (Macherey Nagel, Dueren, Germany).

RNA concentration was measured by absorbance at 260 nm and controlled by optical density ratio at 260/280 nm (1.8 < ratio < 2) and 260/230 nm (2 < ratio < 2.2).

Reverse transcription of total RNA into complementary deoxyribonucleic acid (cDNA) was performed with the cDNA synthesis kit Protoscript First Strand cDNA Synthesis Kit^®^ (Biolabs E6560, New England Biolabs, Ipswich, MA, USA). Quantitative RT-PCR was achieved with a SYBR^®^ Green PCR kit (Applied Biosystems, Foster City, CA, USA) in a QuantStudioTM 7 Pro system (Applied Biosystems, Life Technologies Ltd., Singapore) under the following cycling conditions: 2 min at 50 °C; 10 min at 95 °C; 40 cycles of 15 s at 95 °C and 1 min at 60 °C; and a final dissociation step.

The primer sequences used in this experiment are listed in Table 1.

The mRNA levels were normalized using the transcripts of the 18S and GAPDH housekeeping gene. The geometric mean Ct of GAPDH and 18S was used to normalize the results. Design and Analysis software v2.6.0 (Applied Biosystems, Life Technologies Ltd., Singapore) was used for quantitative results determination. Each gene analysis was carried out in triplicate. The recommendations of Taylor and Mrkusich were used for primers validation, purity and RNA integrity, and amplification efficiency [31].

#### 2.3.2. Alcian Blue Staining

On day 7, hDPSC and hPDLSC were fixed in 4% buffered formaldehyde (Sigma-Aldrich, Saint-Louis, MO, USA) at 1-h RT. The fixed cells were concentrated by centrifugation, excess buffer formaldehyde was removed, and then the cells were washed three times with PBS to remove excess fluid.

Then, two drops of Reagents were applied (Thermo Scientific Shandon Cytoblock Cell Block), and cells were embedded in paraffin wax with Excelsior ES 50 for 3 h.

Paraffin blocks sections were cut at 4µm thickness and mounted on glass slides.

After deparaffinization, cells sections were stained with Alcian Blue pH 2.5 (Leica, Nanterre, France) for 20 min and counterstained with Nuclear Fast Red (Diapath, Martinengo, BG, Italy) for 2 min with Leica ASP 300.

Cell colorations were observed with scanner Hamamatsu Photonics France (Nanozoomer 2.0RS/ C10730-12, Massy, France).

### 2.4. Statistical Analysis

Data analysis was performed with GraphPad Prism v9.3.1 (GraphPad Software, San Diego, CA, USA). The tests used were non-parametric Kruskal–Wallis tests and Dunn’s post hoc tests. Data are expressed as the means ± standard deviation, and differences were considered significant when *p* < 0.05.

## 3. Results

### 3.1. Cell Lines Isolation and Characterization

Before starting the experiments, it was confirmed that the hDPSC isolated from explant cultures had MSCs-like characteristics. They exhibited a spindle appearance, similar to those of MSCs, and they were plastic-adherent cells.

The cytometric flow analysis revealed the expression of CD73, CD90, and CD105, and no expression of CD34 and CD45.

The hDPSC were able to form colonies 10 days after seeding (Figure 2A). Moreover, the osteoblastic and adipogenic differentiation potentials were confirmed by Alizarin Red staining and Oil Red O staining, respectively.

In fact, the hDPSC produced nodules of mineralization after being cultured with osteogenic medium for 21 days (Figure 2B). Similarly, after 21 days of culture with adipogenic medium, the formation of lipid droplets was observed (Figure 2C).

### 3.2. Chondrogenic Differentiation

#### 3.2.1. Alcian Blue Staining

On day 7, Alcian blue staining was positive in all cultures, except for the monolayer with complete medium. The pellet and spheroids cultures were positive for Alcian blue staining, regardless of the culture medium (Figure 3 and Figure 4).

For spheroids cultures on day 7, there were two parameters:Changes in cell shape;Noticeable presence of cartilage extracellular matrix (Figure 3 and Figure 4; arrow).

These changes were more discreet, even absent, in pellets cultures.

#### 3.2.2. Real-Time Polymerase Chain Reaction (RT-qPCR)

The most striking results were the significant increase in Acan and Comp expression in the spheroids of hDPSC cultured with chondrogenic medium compared with those cultured with complete medium, on days 3 and 7.

The same significant increase was observed in the hPDLSC spheroids cultured with chondrogenic medium, except in Acan on day 3 (Figure 5A,B).

In the spheroids with chondrogenic medium, Comp expression was very important, with 128- to 1536-fold increases over complete medium (Table 2 and Table 3).

Acan expression was also very important in the hDPSC spheroids cultured with chondrogenic medium, with 38- to 57-fold increases over complete medium (Table 2).

Regarding pellet culture, Acan expression was increased in the chondrogenic medium only in hDPSC, but the difference was not statistically significant. Comp expression was significantly increased in chondrogenic medium, except for hPDLSC at day 7, where the increase, although not significant, was much lower than in the spheroids (Figure 5A,B, Table 2 and Table 3).

Regarding the monolayer culture, Acan expression was significantly increased in chondrogenic medium compared to complete medium, in both cell lines at days 3 and 7. Acan expression in monolayer with chondrogenic medium was higher in hDPSC compared to hPDLSC, but was much lower than that in the hDPSC spheroids cultured with chondrogenic medium.

Comp expression was significantly increased in the chondrogenic medium only in hDPSC at day 3 and in both cell lines at day 7. Comp expression in the chondrogenic medium was much lower in monolayer compared to the spheroids in both cell lines at days 3 and 7 (Figure 5A,B, Table 2 and Table 3).

At both times, Col2a1 and Sox9 expression was higher in three-dimensional cultures (spheroids and pellets), especially in spheroids, regardless of the culture medium (Figure 5C,D, Table 2 and Table 3).

When comparing the expression of chondrogenic markers in spheroids and pellets cultured with chondrogenic medium to that of the monolayer cultured with chondrogenic medium, the spheroids of hDPSC exhibited increased expression for all markers at both times (Figure 6). The pellets of hDPSC also showed increased expression, except for Acan. In hPDLSC, the increases in expression were lower and did not involve all markers (Figure 6).

## 4. Discussion

The clinical use of MSCs is an attractive therapeutic option, especially due to their multilineage potential and their immunomodulatory properties [32]. MSCs are investigated in many fields, including chronic inflammatory bowel disease [33]; autoimmune, neurodegenerative, and cardiovascular diseases [34]; and tissue engineering [19]. Scaffolds seeded with MSCs are widely studied in bone and cartilage tissue engineering [32]. However, the number of approved MSCs treatments worldwide remains limited [34].

MSCs therapy in OA would be appropriate because of their differentiation capacity and paracrine properties [35]. MSCs clinical trials that have been conducted for OA have used different injection methods: intra-articular injection, combination with a scaffold, or combination with platelet-poor plasma [34]. MSCs are multipotential cells with a high capacity for proliferation and multi-lineage differentiation, such as osteoblasts, chondrocytes, and adipocytes [19]. In the present study, a phenotypic characterization (flow cytometry) of dental MSCs was performed, and their ability to differentiate into osteoblasts and adipocytes was confirmed. Besides, since MSCs and fibroblasts are both plastic adherent, share similar cell morphology, and express many similar cell surface proteins [36]; the presence of MSCs rather than fibroblast was verified by the Colony-Forming Assay [37]. Even if bone marrow was the first source reported to contain MSCs, MSCs can be isolated from various tissues, including adipose tissue, umbilical cord, skeletal muscle, cartilage, synovium, cardiac tissue, and dental tissue [18,19,20]. Dental MSCs exhibit many advantages, with the main one being their very easy access [18,20]. Besides, they can be found in various dental tissues [18]. This ease of access makes them very attractive, including for cartilage regeneration, especially since they were already studied on OA models and showed promising results [38,39]. These advantages led us to choose dental MSCs to conduct the present study. Furthermore, since the tissue source can vary the properties of human MSCs [40], we chose to use two cell lines from two different parts of the same organ: dental pulp and periodontal ligament. Indeed, even if MSCs have a dental origin, they have shown different properties depending on their source (dental pulp, periodontal ligament, apical papilla, dental follicle, etc.) [41], which is confirmed by the present study, with hDPSC and hPDLSC exhibiting different gene expression profiles. We chose hDPSC and hPDLSC since both have been reported to differentiate into chondrocyte [13,42]. Besides, both showed a higher number of population doublings than bone marrow MSCs in culture [43] and can be easily isolated from almost all adults [42,44]. Moreover, although dental pulp and periodontal ligament have immunologic, neurologic, pathologic, and circulatory similarities, they have different functions [44].

Our MSC populations were characterized in accordance with the recommendations of the International Society for Cellular Therapy (plastic adherence, expression of the surface markers, lacking the expression of hematopoietic markers, capacity of in vitro differentiation into adipocyte and osteoblast lineages, and fibroblast-like spindle shape in culture) [19,45].

Three-dimensional and high-density cell cultures are needed for chondrogenic differentiation [26] because high cell density cultures mimic the condensation of mesenchymal cells that induces chondrogenesis during development [46]. Most often, chondrogenic differentiation is performed with pellet culture, which is the gold standard to maintain chondrocytes in a differentiated stage [26]. Three-dimensional cultures (spheroids or pellets) have many advantages: when cultured in three dimensions, MSCs maintain their intrinsic phenotypic properties by cell–extracellular matrix interactions [47]. Thus, these culture methods are regarded as more physiological, and better mimic in vivo conditions [48]. Besides, 3D cultures result in a higher expression of paracrine factors, including exosomes [49]. Nevertheless, the spheroid culture system has many advantages over pellet culture, including fewer required cells (5 × 10^3^ cells for one spheroid; 1.25 × 10^5^ cells for one pellet) and ease of handling. Besides, recent studies found that the reduction in pellet size can improve chondrogenesis [50]. At last, the use of spheroids for MSCs transplantation is well documented in tissue engineering, and they have been reported to enhance the overall therapeutic potential of MSCs after transplantation [51], whereas few studies have reported the use of pellets. These advantages led us to choose the spheroid culture system for the present study. A comparison was made with pellet culture. Monolayer culture, with complete and chondrogenic media, was used as negative control. The chondrogenesis of dental pulp stem cells cultured as spheroids has been studied by several authors [52,53,54], but few studies have compared spheroid culture to pellet culture (the gold standard for chondrogenic differentiation). Besides, the methods used for spheroid culture differ in the literature; therefore, one of the objectives of this study was to test the liquid overlay technique.

To investigate chondrogenic differentiation, chondrocyte-specific markers expression was assessed by RT-qPCR. The following markers were evaluated, according to the literature [55]:Aggrecan (Acan), a proteoglycan of the extracellular matrix (ECM), which is the major component of cartilage [56].Cartilage oligomeric matrix protein (Comp), which is a non-collagenous extracellular matrix glycoprotein that is primarily found in the human skeleton system (articular cartilage, meniscus, ligaments, tendons, and synovium) [57].Type II collagen (Col2a1), which is one of the main components of the ECM of hyaline articular cartilage [58].SRY-related HMG box-containing-9 (Sox9), which is a master transcription factor that regulates multiple events in chondrogenesis. SOX9 is involved in the transactivation of Col2a1 and Acan [58].

Regarding Acan and Comp markers, their expression was significantly increased when hDPSC were grown as spheroids in chondrogenic medium compared to complete medium, at days 3 and day 7. In hPDLSC, the same results were observed, except for Acan, which was significantly increased only at day 7. Acan was also increased when cells were grown as spheroids in chondrogenic medium compared to complete medium at day 3, but the difference was not significant. This significant increase in expression was very strong for Comp, in both cell lines, and for Acan only in hDPSC. Comp expression was also increased in pellets culture with chondrogenic medium compared with complete medium, at day 3 and day 7 in hDPSC, and only at day 3 in hPDLSC. Nevertheless, its expression was lower than in spheroids, and no increase in Acan expression was observed in pellets. These results show a great potential of spheroid culture for chondrogenic differentiation. The same phenomenon was observed in monolayer, even though the increase in Comp expression was much lower, confirming a great potential of dental MSCs in chondrogenic differentiation.

The Alcian Blue staining results also suggest a great potential of dental MSCs in chondrogenic differentiation, since they were slightly positive in monolayer with chondrogenic medium. Besides, they were also positive in pellet and spheroid cultures in chondrogenic medium, and also in complete medium. Nevertheless, for the 3D culture models (spheroids and pellets), only the spheroids showed changes in cell shape, and the cartilage extracellular matrix clearly appeared. These changes were less obvious for the pellets for this 7-day incubation time. This work will be continued by a quantification of the extracellular matrix using morphometric analysis.

Pellet culture was not the most favorable model for these two cell types for chondrogenic differentiation. In addition, hDPSCs showed better chondrogenic differentiation than hPDLSCs, confirming that dental MSCs have different properties depending on their source [41]. The other markers (Col2a1 and Sox9) did not increase with chondrogenic medium but seemed to increase in three-dimensional cultures (spheroids and pellets), highlighting the need for three-dimensional culture for chondrogenic differentiation. These results are consistent with some studies in the literature: Prideaux et al. developed two multipotent mesenchymal progenitor murine cell lines. In the first one, they found an increase in all chondrogenic markers (Col2a1, Acan, Comp, and Sox9). In the second cell line, only Comp and Sox9 were increased, confirming the results of the present study, in which differences were observed between the cell lines. Besides, as in this study, Prideaux et al. found a very large increase in Comp in cells cultured with chondrogenic medium compared to those cultured in complete medium (about 200-fold) [55]. These results agree with some studies in the literature: in a study by Joanna L. James et al., no increase in Sox9 was observed during chondrogenesis [23]. Similarly, Prideaux et al. generated two cell lines capable of chondrogenic differentiation. One showed an increase in Col2a1 during chondrogenesis, but not the other [55].

In the present study, the Comp marker was strongly increased when cells were cultured as spheroids with chondrogenic medium, compared to all other conditions, at day 3. These results demonstrate that spheroid culture is an excellent model for the chondrogenic differentiation of dental MSCs. In addition to Comp, Acan expression was strongly increased after day 3 in hDPSC spheroids with chondrogenic medium, showing the value of this cell line in chondrogenic differentiation. The results of the present study are consistent with those of Liangming Zhang et al., who found that the spheroid culture system was a promising tool for in vitro chondrogenic studies. In their study, chondrogenic differentiation was investigated at days 7, 14, and 21 [59]. The present study showed that the spheroids of dental MSCs are a comparable or even superior model to the gold standard pellet culture. Although the majority of studies on chondrogenesis showed an optimal result after three or four weeks of culture, the present work demonstrated that, after three and seven days, several markers of chondrocyte differentiation fluctuated. These gene variations are early assessment tools for our three-dimensional chondrocyte cultures and can be used for innovative bioactive glasses and to evaluate our cartilage tissue engineering protocols. The results of the present study were consistent with those of Ji-Yun Ko et al., who showed chondrogenic differentiation as early as day 3 on adipose-derived stromal/stem cells grown in spheroids [60].

It is well known that tissue source can vary the exact properties of human MSCs [40]. In the present study, hDPSC showed better results for chondrogenic differentiation, compared to hPDLSC. Moreover, hDPSC spheroids cultured with chondrogenic medium were the best condition for chondrogenic differentiation. In fact, they were the only ones that showed an increased expression of all markers compared to the negative control. Thus, these results showed that, although derived from the same organ, these two cell lines exhibited different properties. Besides, it can be concluded that hDPSC are the most interesting cell line to develop a chondrocyte model for bioactive glass screening in tissue engineering. This work should be continued in order to confirm the value of dental MSC spheroids in chondrogenic differentiation, and then to study their interest in cartilage tissue engineering.

## 5. Conclusions

The present study showed strong potential for the chondrogenic differentiation of dental MSCs from the third day of culture.

The spheroid culture system using the liquid overlay technique is an excellent model to obtain spheroids of homogeneous size for chondrogenic differentiation.

The very easy access and the multiple sources available make dental MSCs attractive for chondrogenic differentiation.

Dental MSCs exhibit different properties according to their tissue source. In this study, hDPSC showed better chondrogenic differentiation than hPDLSC.

Spheroids of hDPSC would allow for rapidly obtaining chondrocytes with easily accessible cells.

## Figures and Tables

**Figure 1 biomedicines-11-01314-f001:**
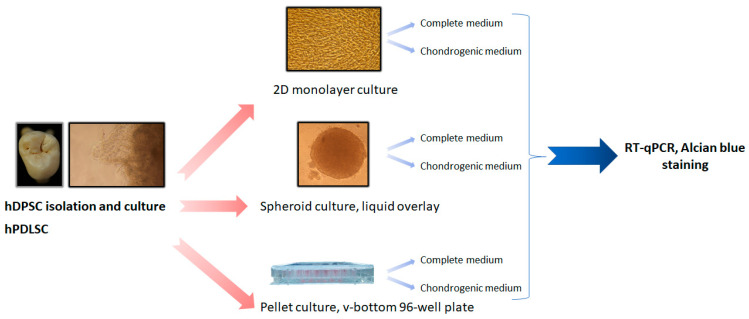
General plan of the study for chondrogenic differentiation.

**Figure 2 biomedicines-11-01314-f002:**
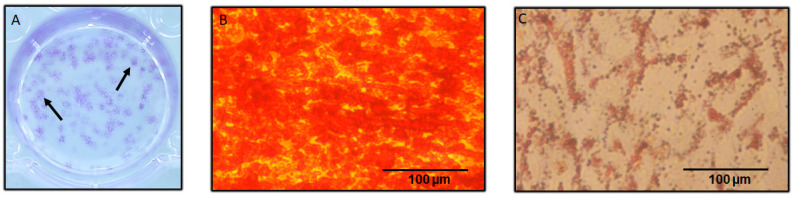
Characterization of hDPSC. (**A**) Colony-forming assay. (**B**) Alizarin red staining. (**C**) Oil Red O staining.

**Figure 3 biomedicines-11-01314-f003:**
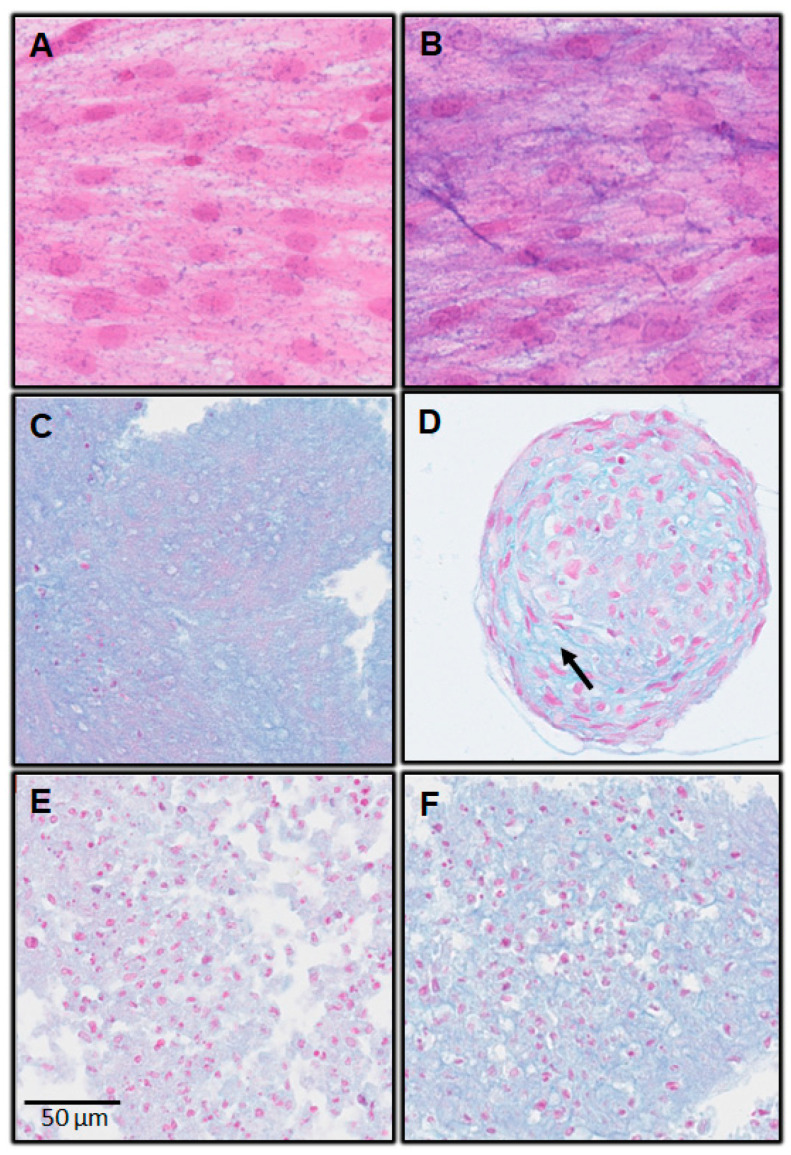
Alcian Blue Staining; hDPSC; day 7. (**A**) Monolayer with complete medium; (**B**) monolayer with chondrogenic medium; (**C**) spheroids with complete medium; (**D**) spheroids with chondrogenic medium; (**E**) pellets with complete medium; (**F**) pellets with chondrogenic medium.

**Figure 4 biomedicines-11-01314-f004:**
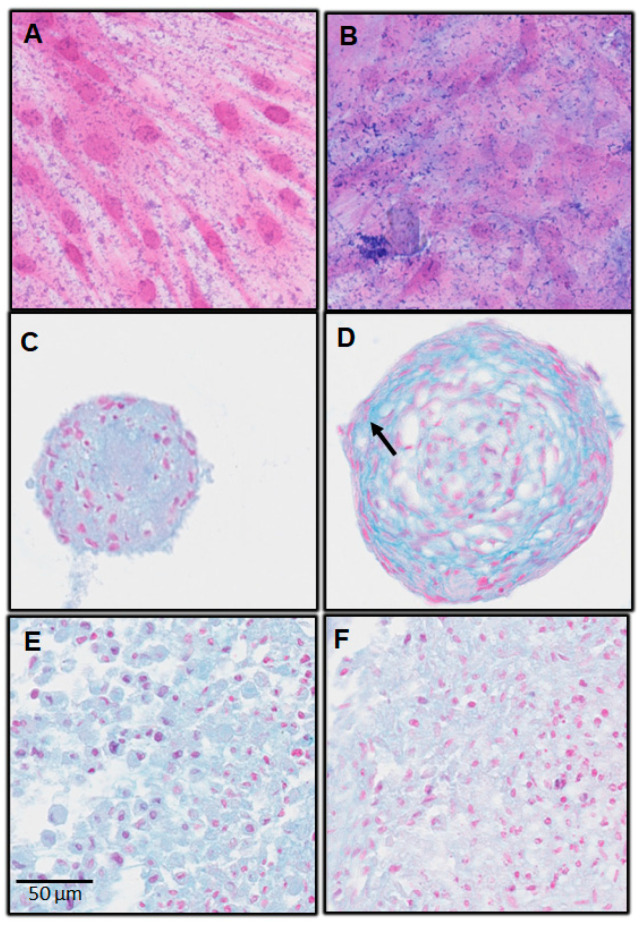
Alcian Blue Staining; hPDLSC; day 7. (**A**) Monolayer with complete medium; (**B**) monolayer with chondrogenic medium; (**C**) spheroids with complete medium; (**D**) spheroids with chondrogenic medium; (**E**) pellets with complete medium; (**F**) pellets with chondrogenic medium.

**Figure 5 biomedicines-11-01314-f005:**
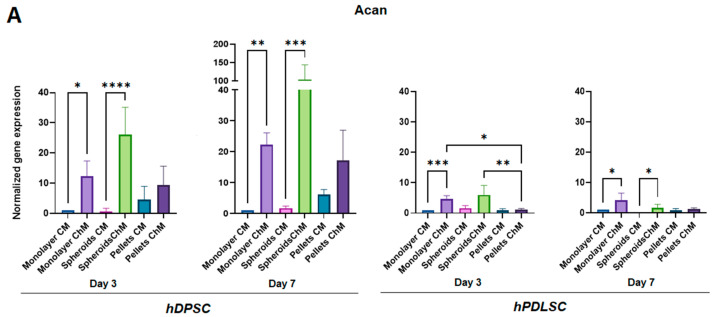
Quantitative reverse transcription-polymerase chain reaction showing gene expression under the different culture conditions. (**A**) Acan; (**B**) Comp; (**C**) Col2a1; (**D**) Sox9. CM: Complete medium; ChM: Chondrogenic medium. *: *p* < 0.05; **: *p* < 0.01; ***: *p* < 0.001; ****: *p* < 0.0001.

**Figure 6 biomedicines-11-01314-f006:**
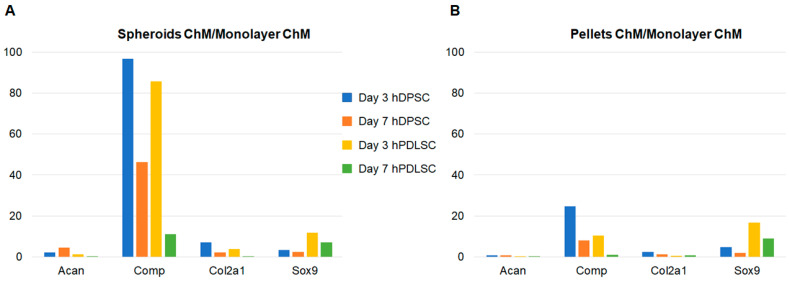
Gene expression relative to monolayer with chondrogenic medium. (**A**) Spheroids with chondrogenic medium; (**B**) pellets with chondrogenic medium.

**Table 1 biomedicines-11-01314-t001:** Description and characteristics of primers used for RT-qPCR.

Primer	Sequence 5′-3′	Exon Position	Product Size (bp)	Primer Efficiency, *E*_p_ (%)	*R* ^2^
18S	F: ATTAAGGGTGTGGGCCGAAG R: GGTGATCACACGTTCCACCT	F: E1/2 R: E2/3	111	110.1	1
GAPDH	F: AAGGTGAAGGTCGGAGTCAAC R: GGGGTCATTGATGGCAACA	F: E2 R: E3	102	90.4	1
ACAN	F: GCACAGCCACCACCTACAAAC R: AGCGACAAGAAGAGGACACCG	F: E15/16 R: E16	175	101.4	0.91
COL2A1	F: GGCAATAGCAGGTTCACGTACA R: CGATAACAGTCTTGCCCCACTT	F: E52 R: E53	79	113.8	0.93
SOX9	F: GAAGCTCGCGGACCAGTA R: TCTCGCTCTCGTTCAGAAGT	F: E1 R: E2	89	96	0.95
COMP	F: AGGGTACCCAACTCAGACCA R: AGTTGTCCCGAGAGTCCTGA	F: E11 R: E13	178	93	1

Abbreviations: GAPDH, glyceraldehyde-3-phosphate dehydrogenase; ACAN, aggrecan; COL, collagen; SOX, SRY-related HMG box-containing; COMP, Cartilage oligomeric matrix protein.

**Table 2 biomedicines-11-01314-t002:** Gene expression under the different hDPSC culture conditions: ratio chondrogenic medium/complete medium.

		Monolayer	Spheroids	Pellets
Day 3	Acan	12.36	38.49	2.03
	Comp	14.72	128.02	293.75
	Col2a1	1.32	0.43	0.39
	Sox9	0.54	1.07	0.79
Day 7	Acan	22.31	57.71	2.79
	Comp	513.74	910.12	1046.70
	Col2a1	2.51	0.57	1.19
	Sox9	0.55	0.68	0.62

**Table 3 biomedicines-11-01314-t003:** Gene expression under the different hPDLSC culture conditions: ratio chondrogenic medium/complete medium.

		Monolayer	Spheroids	Pellets
Day 3	Acan	4.73	3.64	1.27
	Comp	25.26	1536.67	278.17
	Col2a1	1.20	0.33	0.28
	Sox9	0.53	0.96	1.22
Day 7	Acan	4.24	59.33	1.40
	Comp	2122.08	1144.45	257.64
	Col2a1	4.28	0.17	0.56
	Sox9	0.40	0.63	0.74

## Data Availability

Not applicable.

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
