# Peer review of "Spheroid Culture System, a Promising Method for Chondrogenic Differentiation of Dental Mesenchymal Stem Cells"

_biomedicines, 2023, doi:10.3390/biomedicines11051314_

Round 1

Reviewer 1 Report

Dear Authors, 

I read your article: "Spheroid culture system, a promising method for chondrogenic differentiation of dental mesenchymal stem cells" and I find it very interesting.

However, I suggest some things to add to improve and increase the importance of your work.

-        line  48 improve references about other MSCs lineages differentiation (DOI: 10.1080/19336896.2018.1463797DOI: 10.1016/j.yexcr.2015.11.012DOI: 10.3390/antiox10050716).

-        line 55 add references about MSCs isolation from Dental pulp (for example doi: 10.1073/pnas.240309797; DOI10.3791/59282).

-        In discussion describe other therapeutic application for MSCs

-        Review the References according to the instructions for the authors: https://www.mdpi.com/journal/ijms/instructions

Author Response

Response to Reviewer 1 Comments

Dear Authors,  

I read your article: "Spheroid culture system, a promising method for chondrogenic differentiation of dental mesenchymal stem cells" and I find it very interesting.

However, I suggest some things to add to improve and increase the importance of your work.

-        line  48 improve references about other MSCs lineages differentiation (DOI: 10.1080/19336896.2018.1463797; DOI: 10.1016/j.yexcr.2015.11.012; DOI: 10.3390/antiox10050716).

-        line 55 add references about MSCs isolation from Dental pulp (for example doi: 10.1073/pnas.240309797; DOI10.3791/59282).

-        In discussion describe other therapeutic application for MSCs

-        Review the References according to the instructions for the authors: https://www.mdpi.com/journal/ijms/instructions

Response:

Dear Reviewer,

I thank you for the interest you have shown in our work.

We added the recommended references, and we described other therapeutic application for MSCs in the discussion.

We have also reviewed our References.

Reviewer 2 Report

The authors compared the capacity of chondrogenic differentiation of dental pulp MSCs and periodontal ligament MSCs in the form of spheroids. The following comments are provided for the authors’ consideration:

1.     Lack of novelty, many papers had reported the chondrogenesis of DPSCs in the form of spheroid:

J. Clin. Med. 2020, 9(1), 242; https://doi.org/10.3390/jcm9010242;

Stem Cell Rev and Rep 17, 1810–1826 (2021). https://doi.org/10.1007/s12015-021-10172-4;

Bull Exp Biol Med 170, 528–536 (2021). https://doi.org/10.1007/s10517-021-05101-x

2.     The comparison between DPSCs and PDLSCs in the form the spheroid for chondrogenesis is a good point because I didn’t see paper reporting this comparison. There are lots of paper compared these two kinds of stem cells from various perspectives,

https://doi.org/10.1016/j.joen.2021.11.005; https://doi.org/10.1371/journal.pone.0071101; https://doi.org/10.1089/scd.2009.0446.

However, in this manuscript, the authors didn’t explain clearly why the current study is necessary.

3.     The whole manuscript was poorly written. For example, in the introduction, the authors didn’t provide enough background, as mentioned in point 1, lots of papers had reported chondrogenic spheroids, where is the research gap that this study is aimed to fill; the discussion is unfocused, and why the discussion were divided into so many paragraphs.

4.     The experimental design is not convincing. As we know chondrogenic differentiation usually take 3-4 weeks to observe. Here the authors used 3 and 7 days, so from Figure 3 and 4, I didn’t see difference on the blue colour between the groups in complete medium and  the groups in chondrogenic medium, which means the chondrogenesis is not evident. Besides, I don’t understand why the authors set the monolayer groups as we know it is difficult to differentiate monolayer MSCs into chondrocytes. Why the comparison between DPSC and PDLSC  in Figure 6 was conducted using the ratio of spheroid or pellet to monolayer. Why the characteristic data for both cells, such as flow cytometry, were not provided. Only multi-differentiation of DPSC was shown in Figure 2. And the Oil red staining looks blurred and I cannot see red oil drops.

Author Response

Response to Reviewer 2 Comments

The authors compared the capacity of chondrogenic differentiation of dental pulp MSCs and periodontal ligament MSCs in the form of spheroids. The following comments are provided for the authors’ consideration:

  1. Lack of novelty, many papers had reported the chondrogenesis of DPSCs in the form of spheroid:
  2. Clin. Med.2020, 9(1), 242; https://doi.org/10.3390/jcm9010242;

Stem Cell Rev and Rep 17, 1810–1826 (2021). https://doi.org/10.1007/s12015-021-10172-4;

Bull Exp Biol Med 170, 528–536 (2021). https://doi.org/10.1007/s10517-021-05101-x

Response 1:

Dear Reviewer,

I thank you for your comments.

Indeed, many articles have reported chondrogenesis of DPSCs in the form of spheroids, but few studies have compared spheroid culture to pellet culture (the gold standard for chondrogenic differentiation). Besides, the methods used for spheroid culture differ in the literature, so one of the objectives of this study was to test the liquid overlay technique. We have added explanations to this in the discussion.

  1. The comparison between DPSCs and PDLSCs in the form the spheroid for chondrogenesis is a good point because I didn’t see paper reporting this comparison. There are lots of paper compared these two kinds of stem cells from various perspectives,

https://doi.org/10.1016/j.joen.2021.11.005; https://doi.org/10.1371/journal.pone.0071101; https://doi.org/10.1089/scd.2009.0446.

However, in this manuscript, the authors didn’t explain clearly why the current study is necessary.

Response 2:

One of the objectives of this work was to use two cell lines from two different parts of the same organ, as the literature indicates that dental stem cells have different biological properties depending on their source. This is confirmed by the present study, with hDPSCs and hPDLSCs showing different gene expression profiles. We have added explanations to this in the discussion.

  1. The whole manuscript was poorly written. For example, in the introduction, the authors didn’t provide enough background, as mentioned in point 1, lots of papers had reported chondrogenic spheroids, where is the research gap that this study is aimed to fill; the discussion is unfocused, and why the discussion were divided into so many paragraphs.

Response 3:

The aim of this work is to set up a quick test for the bioactive glass screening for bone and cartilage tissue engineering (which is the thematic of our laboratory). This study is the first step of this work. We have added explanations about this in the introduction and in the discussion.

  1. The experimental design is not convincing. As we know chondrogenic differentiation usually take 3-4 weeks to observe. Here the authors used 3 and 7 days, so from Figure 3 and 4, I didn’t see difference on the blue colour between the groups in complete medium and  the groups in chondrogenic medium, which means the chondrogenesis is not evident. Besides, I don’t understand why the authors set the monolayer groups as we know it is difficult to differentiate monolayer MSCs into chondrocytes. Why the comparison between DPSC and PDLSC  in Figure 6 was conducted using the ratio of spheroid or pellet to monolayer. Why the characteristic data for both cells, such as flow cytometry, were not provided. Only multi-differentiation of DPSC was shown in Figure 2. And the Oil red staining looks blurred and I cannot see red oil drops.

Response 4:

Indeed, most studies use 3 to 4 weeks of culture to obtain the most advanced chondrogenesis possible. But to get a screening test, a quick test was needed. Ji-Yun Ko et al. showed that differences appear in a very short period of time. We based our model on this work (Ko J, Park J, Kim J, Im G. Characterization of adiposederived stromal/stem cell spheroids versus singlecell suspension in cell survival and arrest of osteoarthritis progression. J Biomed Mater Res A. juin 2021;109(6):86978).

Our study clearly showed that for the gold standard and our model, there was a variation in the expression of markers as early as day 3, whatever the cell line used. As these markers are relevant for chondrogenesis, it did not seem appropriate to wait 3-4 weeks for optimal differentiation.

We have added explanations about this in the discussion.

Besides, we added explanations on blue Alcian staining, and on monolayer which was our negative control to assess the fluctuation of markers under chondrogenic medium.

Consistent with work reported in the literature, there was little fluctuation in markers in monolayer. That’s why we did the figure 6, to express results according to the basal expression of the monolayer.

The characteristic data for hPDLSC were not provided because their characterization was done in a previous work (Novello S, Tricot-Doleux S, Novella A, Pellen-Mussi P, Jeanne S. Influence of Periodontal Ligament Stem Cell-Derived Conditioned Medium on Osteoblasts. Pharmaceutics. 28 mars 2022;14(4):729). Therefore, we only included results from the newly isolated cell line (hDPSC).

We changed the illustration for Red Oil staining.

Reviewer 3 Report

In this study, the authors tried to develop an effective strategy to differentiate dental MSCs into chondrocytes, using a tridimensional cell culture as spheroids comapred with the methods for culture with cell pellets and monolayer culture. The chondrogenic differentiation potential of dental pulp and periodontal ligament MSCs was assessed by RT-qPCR and Alcian Blue staining. The results showed a better chondrogenic differentiation in spheroid than in pellet culture from day 3. Dental pulp MSCs showed a better chondrogenic differentiation than periodontal ligament MSCs. This study demonstrated that the culture system with spheroids method showed better chondrogenic differentiation. It is intersting, however, the authors only cite  a reference about spheroids culture, it is not clear for the readers. Please add some description in the method section.     

Author Response

Response to Reviewer 3 Comments

In this study, the authors tried to develop an effective strategy to differentiate dental MSCs into chondrocytes, using a tridimensional cell culture as spheroids comapred with the methods for culture with cell pellets and monolayer culture. The chondrogenic differentiation potential of dental pulp and periodontal ligament MSCs was assessed by RT-qPCR and Alcian Blue staining. The results showed a better chondrogenic differentiation in spheroid than in pellet culture from day 3. Dental pulp MSCs showed a better chondrogenic differentiation than periodontal ligament MSCs. This study demonstrated that the culture system with spheroids method showed better chondrogenic differentiation. It is intersting, however, the authors only cite  a reference about spheroids culture, it is not clear for the readers. Please add some description in the method section.     

Response:

Dear Reviewer,

I thank you for the interest you have shown in our work.

As you recommended, we have described the method for spheroids culture.

Round 2

Reviewer 2 Report

The abstract should summarize the main findings in the manuscript, it could be improved in terms of scientific writing. The words "better" and "interesting" are too general. The use of the bracket is inappropriate.

Author Response

Response to Reviewer 2 Comments

The abstract should summarize the main findings in the manuscript, it could be improved in terms of scientific writing. The words "better" and "interesting" are too general. The use of the bracket is inappropriate

Dear Reviewer,

I thank you for your comments.

We have rewritten the abstract based on your suggestions and in accordance with our previous revisions.

Best regards